# Empirical prediction of variant-activated cryptic splice donors using population-based RNA-Seq data

Ruebena Dawes [1,2], Himanshu Joshi[1] & Sandra T. Cooper [1,2,3 ✉]

Predicting which cryptic-donors may be activated by a splicing variant in patient DNA is notoriously difficult. Through analysis of 5145 cryptic-donors (versus 86,963 decoy-donors not used; any GT or GC), we define an empirical method predicting cryptic-donor activation with 87% sensitivity and 95% specificity. Strength (according to four algorithms) and proximity to the annotated-donor appear important determinants of cryptic-donor activation. However, other factors such as splicing regulatory elements, which are difficult to identify, play an important role and are likely responsible for current prediction inaccuracies. We find that the most frequently recurring natural mis-splicing events at each exon-intron junction, summarised over 40,233 RNA-sequencing samples (40K-RNA), predict with accuracy which cryptic-donor will be activated in rare disease. 40K-RNA provides an accurate, evidence-based method to predict variant-activated cryptic-donors in genetic disorders, assisting pathology consideration of possible consequences of a variant for the encoded protein and RNA diagnostic testing strategies.

[1] Kids Neuroscience Centre, Kids Research, Children's Hospital at Westmead, Sydney NSW2145, Australia. [2] Discipline of Child and Adolescent Health, Faculty of Health and Medicine, University of Sydney, Sydney NSW2006, Australia. [3] The Children's Medical Research Institute, 214 Hawkesbury Road, WestmeadNSW 2145 Sydney, Australia. ✉email: sandra.cooper@sydney.edu.au

Genetic variants affecting the conserved sequences of the consensus splicing motifs can alter binding of spliceosomal components and induce mis-splicing of precursor messenger RNA (pre-mRNA)[1], making them a common cause of inherited disorders[2–5]. Splicing variants can simultaneously induce different mis-splicing outcomes, including skipping of one or more exons, activation of a cryptic splice-site(s), and/or retention of one or more introns[1]. Whether induced mis-splicing disrupts the reading frame or affects a region of known functional (and clinical) importance, has significant diagnostic implications. Therefore, knowing the specific mis-splicing outcome of genetic variant is necessary to conclusively link it to a disease. While the accuracy of in silico algorithms in predicting whether a variant will cause mis-splicing has been comprehensively assessed[6–9], there is currently no reliable means to predict which mis-splicing event(s) may occur in response to a variant that activates mis-splicing. As a result of this and other factors, the vast majority of splice site variants are classified as variants of uncertain significance (VUS); a non-actionable diagnostic endpoint in genomic medicine[10].

We recently evaluated the accuracy and concordance of SpliceAI (SAI)[11] and algorithms within Alamut Visual® (Interactive Biosoftware, Rouen, France)[12,13], to predict splicing outcomes arising from genetic variants identified in 74 families with monogenetic conditions subject to RNA diagnostic studies (79 variants; 19% essential GT-AG splice-site variants and 71% extended splice-site variants)[14]. Algorithmic predictions of the strengths of activated cryptic splice sites were highly discordant, especially for cryptic donors. SAI's deep learning showed the greatest accuracy in predicting activated cryptic splice-site(s) (66% true positive with 34% false negative), whereas historical algorithms within Alamut Visual® resulted in 34–69% false negatives[14].

In this study we focus on determining empirical features that inform prediction of variant-associated spliceosomal selection of a cryptic-donor, in preference to the annotated-donor and other nearby decoy-donors (any GT or GC not used by the spliceosome). Through analysis of 4811 variants in 3399 genes, we show that while splice-site strength and proximity to the annotated-donor strongly influence spliceosomal selection of a cryptic-donor, these factors alone are not sufficient for accurate prediction. Importantly, we show that the most common mis-splicing events seen at each exon-intron junction across 40,233 publicly available RNA-seq samples compiled within the 40K-RNA database, predict with accuracy which cryptic-donor will be activated in rare disease.

## Results
### Reference database of variants activating a cryptic-donor.
We collate a database of cryptic-donor variants, defined as variant-associated erroneous use of a donor other than the annotated-donor. Variants were derived from several sources[11,15,16] (Fig. 1a, see methods). The genomic locations and extended sequences of the annotated-donor, cryptic-donor(s), as well as any decoy-donors (any GT/GC motif within 250 nucleotides (nt) of the annotated-donor), were compiled for analysis. We define the extended donor splice-site region as spanning the fourth to last nucleotide of the exon ($E^{-4}$, E = exon) to the eighth nucleotide of the intron ($D^{+8}$; D = donor), as constraint on sequence diversity eases beyond this point (supplementary Fig. 1).

Cryptic-donor variants fall into three categories (Fig. 1b, Box 1): A) Annotated-Modified (AM): a genetic variant modifies the annotated-donor resulting in activation one or more unmodified cryptic-donors ($n = 2186$) (Fig. 1c–e). AM-variants which are SNVs and DNA insertions commonly affect the $E^{-1}$,

$D^{+1}$, $D^{+2}$ and $D^{+5}$ positions of the annotated-donor (Fig. 1c), and AM-variants which are DNA deletions ranged from 1 to 57 nts in length (Fig. 1d). 89% of AM-variants result in use of a single cryptic-donor, 9% activate 2 cryptic-donors and 2% activate 3 or more cryptic-donors (Fig. 1e).

B) Cryptic-Modified (CM): a genetic variant modifies a cryptic-donor and does not affect the annotated-donor ($n = 2252$) (Fig. 1f, g). CM-variants most frequently affect the $D^{+2}$ position of the cryptic-donor (Fig. 1f), with 32% of all CM SNVs changing the cryptic-donor essential splice motif from GC to GT (Fig. 1g).

C) Annotated-Modified/Cryptic-Modified (AM/CM): a genetic variant that simultaneously modifies the annotated-donor and nearby cryptic-donor ($n = 373$) (Fig. 1h–j). AM/CM-variants which are SNVs and DNA insertions also most frequently affect the $D^{+2}$ position (122/373) of the cryptic-donor (Fig. 1h), with 31% of SNVs converting a GC to GT (Fig. 1i). AM/CM-variants which are DNA deletions range from 1 to 36 nts in length (Fig. 1j).

### 87% of cryptic-donors lie within 250 nt of the annotated-donor.
99% of cryptic-donors activated by AM-variants and 71% of cryptic-donors activated by CM-variants, lie within 250 nt of the annotated-donor (87% collectively, Fig. 2a, b). By definition, AM/CM-variants activate a cryptic-donor that spatially overlaps the annotated-donor; 26% of AM/CM cryptic-donors lie at either the $E^{-4}$ or $D^{+5}$ position (Fig. 2c). For exonic cryptic-donors activated at $E^{-4}$, the GT at $D^{+1/+2}$ of the annotated-donor becomes $D^{+5/+6}$ of the cryptic-donor; conversely for intronic cryptic-donors activated at $D^{+5}$, the GT at $D^{+5/+6}$ of the annotated-donor becomes $D^{+1/+2}$ of the cryptic-donor).

While decoy-donors are present everywhere, which ones are selected as cryptic-donors by the spliceosome in the context of a genetic variant appears strongly influenced by their proximity to the annotated-donor (Fig. 2a, b), as shown by their enrichment at proximal locations relative to all decoys present in the genome (Fig. 2d). The steep decline in exonic decoys (Fig. 2d, left) is due to the shorter lengths of exons limiting their frequency at these distances (50th and 90th percentile for exon length shown). Notably, each annotated-donor has on average 36 decoy-donors within $+/-250$ nt not used by the spliceosome – indicating that features other than proximity to the annotated-donor define a usable cryptic-donor (Fig. 2e).

### Only 31–67% of1 cryptic-donors are stronger than the annotated-donor.
We examined the ability of four algorithmic measures of splice-site strength to predict cryptic-donor activation (Fig. 3). We compared the performance of MaxEntScan (MES)[13], NNSplice (NNS)[12] and SpliceAI (SAI)[11] as well as our own method termed Donor Frequency (DF) (see methods and supplementary Fig. 1 for details, supplementary Fig. 2a–c for full plots). DF measures donor strength based on how many annotated-donors in the human genome have the exact same sequence. DF calculates the median frequency of four consecutive windows of nine nucleotides in length (between $E^{-4}$ and $D^{+8}$) among all annotated-donors, converted to a cumulative frequency distribution. For example, if an $E^{-3}$ to $D^{+6}$ sequence has a raw frequency of 222, this combination of nine bases occurs at the analogous position for 222 annotated-donors, corresponding to the 35th percentile of a cumulative frequency distribution across the human genome (see supplementary Fig. 1c). For these and all further analyses, we excluded the 1113 cryptic variants in the database derived from SAI predictions already validated on GTEx RNA-seq data[11]. Our nomenclature of REF and VAR corresponds to the reference (REF) or variant (VAR) donor sequence.

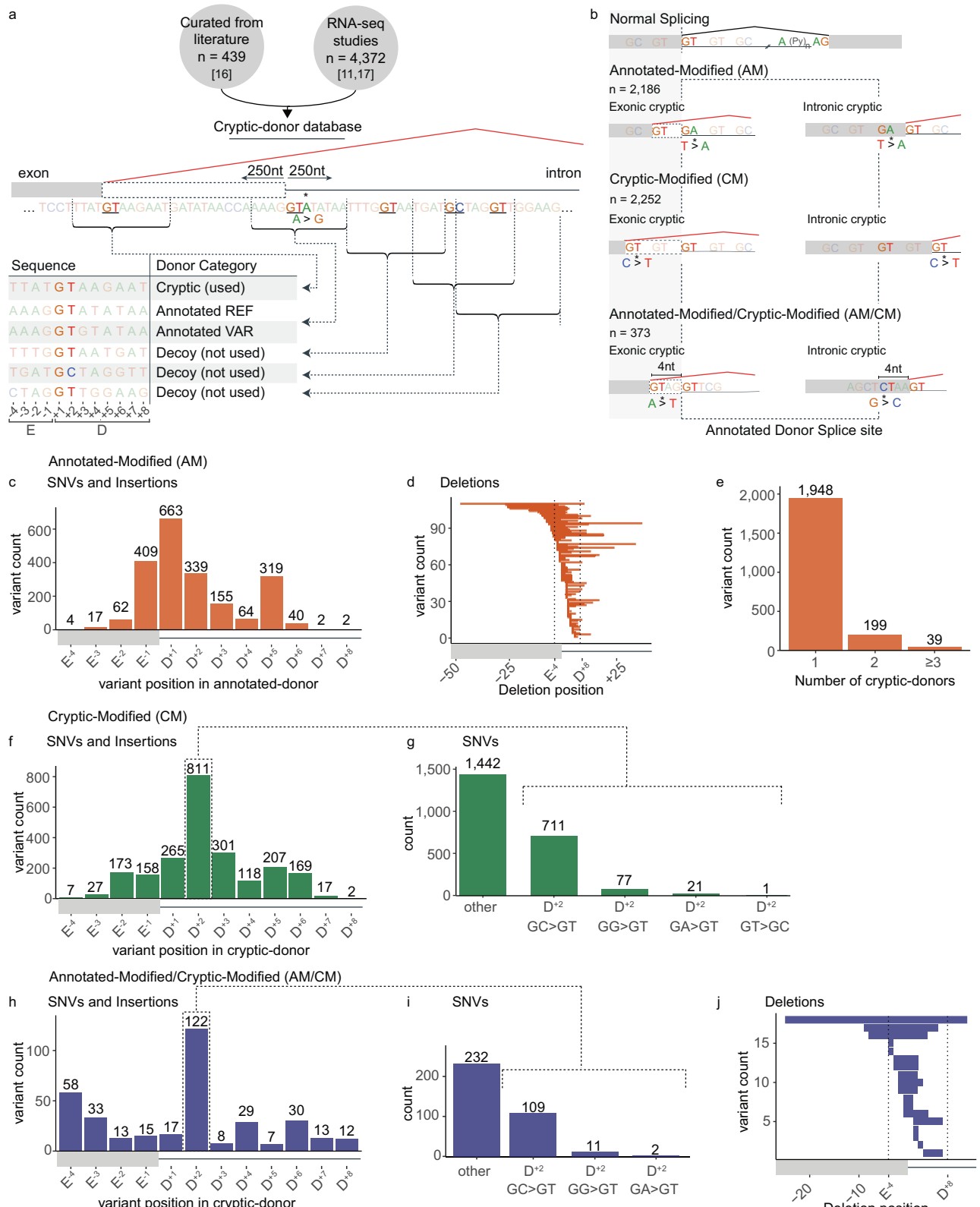

**Fig. 1 Reference database of variants activating a cryptic-donor. a** Schematic of the cryptic-donor database. E = Exon, D = Donor. See methods. **b** Three categories of cryptic-donor variants in the database: Annotated-Modified (AM-variants), Cryptic-Modified (CM-variants) and Annotated-Modified and Cryptic-Modified (AM/CM-variants). **c–e** Characteristics of AM-variants (*n* = 2186, orange). Positions of AM (**c**) Single Nucleotide Variants (SNVs), insertions and (**d**) deletions relative to the annotated-donor. In (**d**) each of the horizontal bars represents one deletion variant showing the position and width of each deletion relative to the annotated-donor. **e** The number of cryptic-donors activated by each AM-variant. **f–g** Characteristics of CM-variants (*n* = 2255; green). **f** Positions of CM SNVs and insertions relative to the cryptic-donor. **g** Frequency of SNVs resulting in cryptic activation, highlighting the prevalence or GC > GT D$^{+2}$ variants. **h–j** Characteristics of AM/CM-variants (*n* = 373, blue). **h**, **i** As in **f**, **g**. **j** As in **d**.

---

**Box 1 | Glossary**

*Annotated-donor*: A donor in an ensembl-annotated transcript.
*Decoy-donor*: Any essential donor dinucleotide (GT/GC) that is not an annotated-donor.
*Cryptic-donor*: A decoy-donor shown to be activated (i.e. used by the spliceosome) by a genetic variant.
*Annotated-Modified (AM)*: A genetic variant modifies the annotated-donor resulting in activation one or more unmodified cryptic-donors.
*Cryptic-Modified (CM)*: A genetic variant modifies a cryptic-donor and does not affect the annotated-donor.
*Annotated-Modified/Cryptic-Modified (AM/CM)*: A genetic variant that simultaneously modifies the annotated-donor and nearby cryptic-donor.
*Donor Frequency (DF)*: A measure of donor strength based on how many annotated-donors in the human genome have the exact same sequence.
*Competitive decoy-donor*: A decoy-donor with a DF score at least 10% the score of the nearby annotated-donor.
*40K-RNA*: An aggregated database of splice-junctions detected across 40,233 publicly available RNA-seq samples.

---

The four algorithms use different methods to measure the intrinsic strength of a given donor splice-site. In the following discussion we use the term stronger and weaker to denote a donor that has a higher or weaker score, respectively, according to that algorithm. Comparisons such as weaker by >50% denote that the donor score has been reduced by more than half by the variant.

For AM-variants, activation of a cryptic-donor typically occurs in the context of a variant that weakens the annotated-donor to less than half of its original strength (Fig. 3a, dark blue). While many AM cryptics are stronger than the annotated$_{VAR}$ (Fig. 3c, example shown in Fig. 3b), a substantial subset are not the strongest decoy-donor within 250nt (Fig. 3d). In fact, many activated cryptic-donors are not recognised as bona fide donors by the respective algorithms, notably NNS (Fig. 3e).

Intuitively, for most CM-variants the cryptic is strengthened by the variant (Fig. 3f, orange, example shown in Fig. 3g). However, less than half of activated cryptics are stronger than the annotated-donor (Fig. 3h). Along similar lines, for a majority of AM/CM-variants the annotated-donor is weakened (Fig. 3i, blue) while the adjacent cryptic is strengthened by the variant (Fig. 3j, orange, example shown in Fig. 3k). However, only 29–67% of AM/CM-cryptics$_{VAR}$ are stronger than the annotated-donor$_{VAR}$ (Fig. 3l). Despite similar overall performance for each algorithm, they showed discordance in variant outcome predictions (Fig. 3M, N) and measures of splice-site strength (Supplementary Fig. 2d).

In summary, four independent algorithms concur that cryptic-donor activation typically occurs in response to weakening of the annotated-donor (85–99% of variants) or strengthening of the cryptic-donor (67–98% of variants). However, only 35–70% of activated cryptic-donors are stronger than the annotated-donor$_{VAR}$, and for unmodified cryptic-donors, 29–62% are not the strongest decoy-donor within 250 nt. Thus, while relative strength of the annotated- and cryptic-donor influence spliceosomal use, there are other factors at play.

**Competitive decoy-donors are depleted close to annotated-donors.** Decoy-donors of comparable or greater strength to the annotated-donor rarely occur within 150 nt (Fig. 4a, red). However, exonic and intronic regions around donors have characteristic single and dinucleotide frequencies which may contribute to the rarity of decoy-donors (supplementary Fig. 3). In particular, the first 50 nt of the intron often shows enrichment in G and T dinucleotides, with distinct patterns: 1) G repeats are enriched in the shortest of introns and T repeats in the longest (supplementary Fig. 3c); 2) Introns with G (or C) at the D$^{+3}$ position are enriched in G dinucleotides whereas introns with A (or T) at the D$^{+3}$ position are enriched in T dinucleotides (supplementary Fig. 3d); 3) Introns with rare donors (low DF) are enriched in T-repeats compared to introns with the most common donors (supplementary Fig. 3e). Therefore, we adapted a previously used method[17] which takes dinucleotide preferences into account, to assess if decoy-donors occur more or less

commonly than expected by random chance (see Methods and supplementary Fig. 4).

GT decoy-donors show increasing exonic depletion approaching the annotated-donor, with out-of-frame decoys (red) depleted more than in-frame decoy-donors (orange), while showing negligible depletion in the intron (Fig. 4b). GC decoy-donors show no depletion in either the exon or the intron (supplementary Fig. 5a).

We next assessed what proportion of decoy-donors in the genome are used, albeit rarely, via unannotated splice-junctions detected across 40,233 publicly available RNA-seq samples from GTEx[18] and Intropolis[19] (40K-RNA). Unannotated splice-junctions (representing stochastic mis-splicing), seen rarely in RNA-seq samples aggregated across a population, constitute empirical evidence that both splicing reactions can be executed using a decoy-donor. Therefore, we mined 40K-RNA for splice-junctions representing the use of cryptic-donors within 250 nt of any annotated-donor, and ranked them according to the number of samples they were present in (see methods). Overall, ~7% of all unannotated decoy-donors are in fact present as rare, stochastic mis-splicing events in 40K-RNA.

The proportion of exonic GT decoy-donors present in 40K-RNA (relative to all decoys) dramatically increases with proximity to the annotated-donor, with intronic decoys showing only a modest change (Fig. 4c). This mirrors depletion patterns (Fig. 4b) and confirms that decoy-donors closer to the annotated-donor are inherently more likely to be used by the spliceosome. Less than 4% of exonic GC decoy-donors are present in 40K-RNA, even very close to the exon/intron junction, in line with their observed lack of depletion (Supplementary Fig. 5b).

The ability of DF to measure donor strength is evidenced by Fig. 4d, e. While there is negligible depletion of decoy-donor sequences that do not exist as a bona fide donor at any exon-intron junction in GRCh37 (DF = 0, grey), there is clear depletion of exonic decoy-donors closer in DF (50–90% DF, mid-blue), or of similar or greater DF ( ≥90% DF, dark blue) (Fig. 4d, left), relative to the annotated-donor. Depletion is even evident for decoy-donors that have DF of only 10% relative to the annotated-donor, and so we define a competitive decoy-donor as one above this threshold. Interestingly, except for the most competitive decoy-donors (≥90% DF; Fig. 4d, right, dark blue), decoy-donors show no depletion in the intron. Concordantly, the proportion of exonic decoy-donors present in 40K-RNA increases with increasing relative DF, and to a lesser extent at the start of the intron (Fig. 4e).

**Why are intronic decoy-donors less likely to be used by the spliceosome?** The fact that intronic decoy-donors are less depleted and less likely to be seen in 40K-RNA (Fig. 4b–e) was initially perplexing, given that cryptic-donors are just as common in the intron as in the exon (Fig. 2a, b). However, we reasoned distinctive nucleotide preferences in the first ~50 nt of the intron

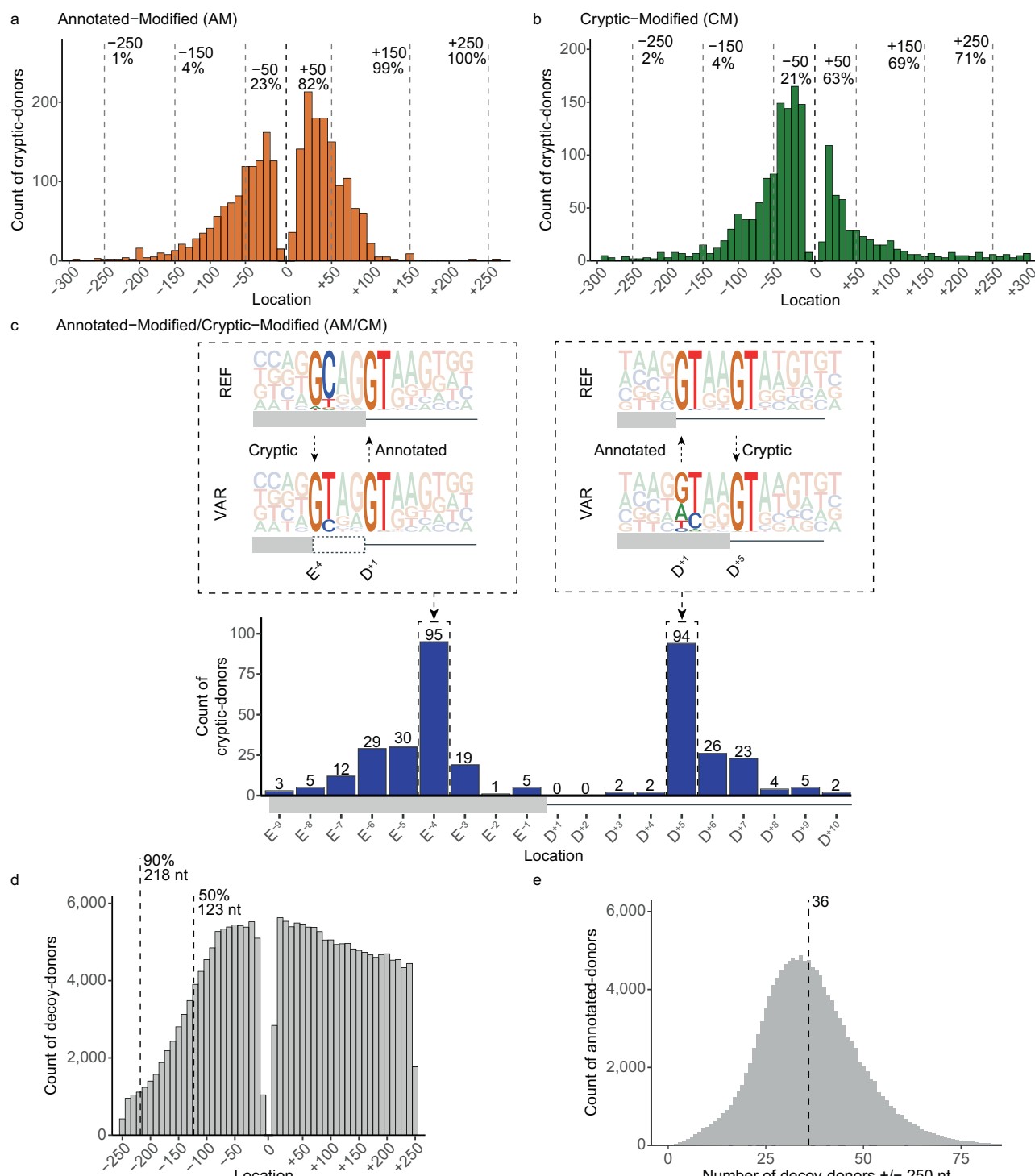

**Fig. 2 Cryptic-donor activation is influenced by proximity to the annotated-donor. a, b** Distribution of cryptic-donors activated by (**a**) AM-variants and (**b**) CM-variants. Location (x-axis) denotes the distance of the cryptic-donor from the annotated-donor, with negative values upstream into the exon and positive values downstream into the intron. **c** (Bottom) Distribution of cryptic-donors activated by AM/CM-variants. (Top) Pictograms showing the Reference (REF) and Variant (VAR) sequences for AM/CM-variants. Activated cryptic-donors are prevalent at $E^{-4}$ (left) and D+5 (right) due to conserved GTs at $D^{+1/+2}$ and $D^{+5/+6}$ of the conserved donor splice-site sequence. **d** Frequency of naturally occurring decoy-donors (any GT or GC) +/− 250 nt of annotated-donors in the human genome. Dashed lines indicate the 50th and 90th percentile for exon length. The decline in exonic donors is due to relatively fewer longer exons. **e** Distribution of the number of decoy-donors in the +/−250 nt surrounding each annotated-donor in the human genome. Dashed line shows that there are an average of 36 decoy-donors within 250 nt of each annotated-donor.

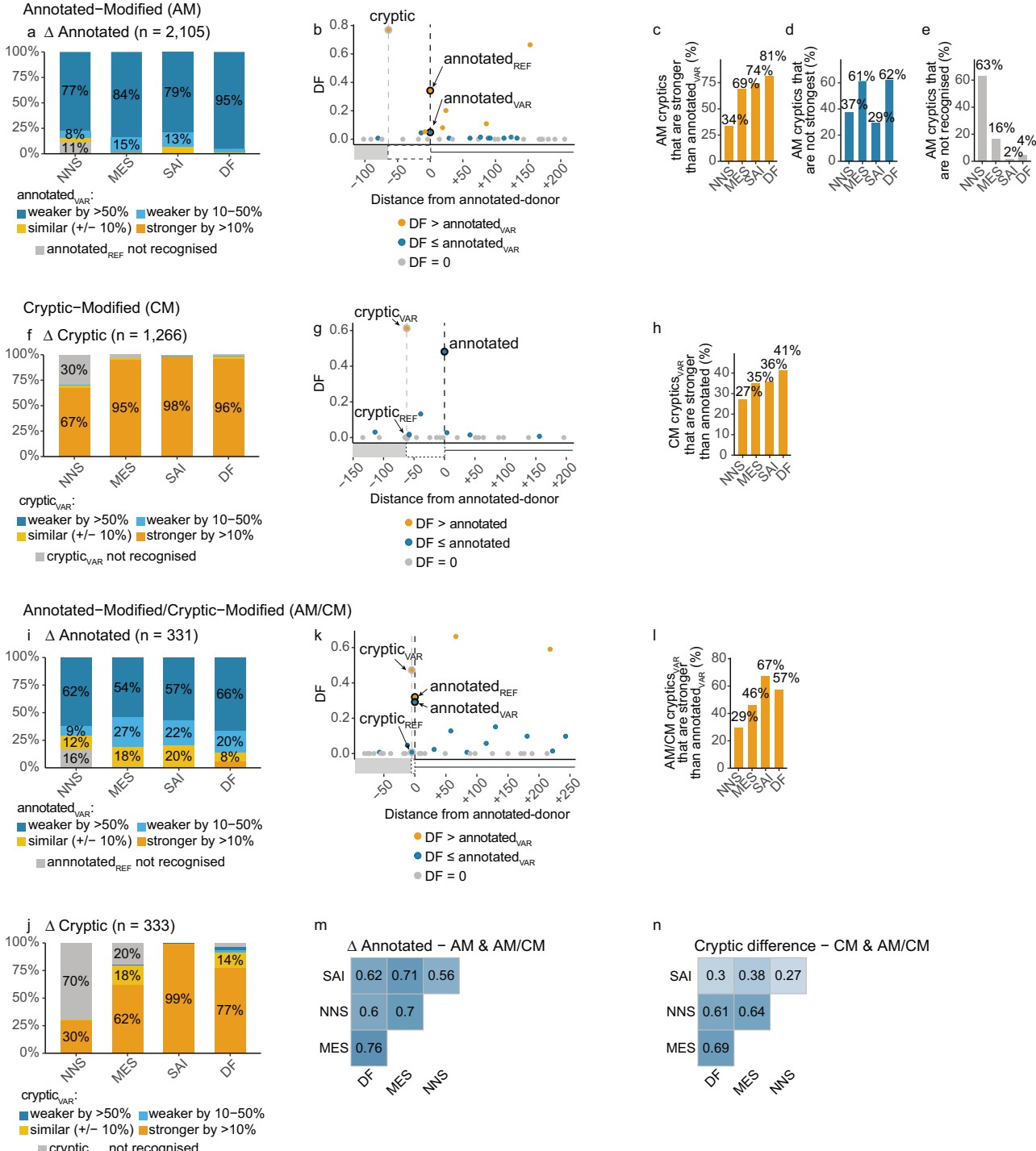

**Fig. 3 Cryptic donor activation is influenced by relative strength. a–e** Assessment of algorithmic scores of splice-site strength for AM-variants. NNS NNSplice, MES MaxEntScan, SAI SpliceAI, DF Donor Frequency. Categories such as weaker by >50% are assigned based on how the score has been impacted by the variant (i.e., more than halved). **a** Proportion of variants with annotated-donor Δ scores (Annotated$_{VAR}$/Annotated$_{REF}$) in each of the categories shown in the figure key. Most AM-variants weaken the annotated-donor by >50% (dark blue). See supplementary Fig. 2 for full plots. **b** Example variant showing the Donor Frequency (DF) scores (see supplementary Fig. 1) for the cryptic-donor (DF = 0.77), versus the reference (REF = 0.34) and variant (VAR = 0.05) annotated-donor, as well as surrounding decoys not used. Vertical dotted lines indicate position of annotated- and cryptic-donors. Donors coloured according to the figure key. **c** Percent of cryptic-donors stronger than the annotated$_{VAR}$. **d** Percent of cryptic-donors that are not the strongest donor splice-site within 250 nt. **e** Percent of AM-variant activated cryptic-donors that are not recognised by each algorithm (i.e., score of 0). **f–h** Strength measures for CM-variants. **f** Proportion of variants with cryptic-donor Δ scores (Cryptic$_{VAR}$/Cryptic$_{REF}$) in each of the categories shown in legend. Most CM-variants strengthen the cryptic-donor by >10% (dark yellow). See supplementary Fig. 2 for full plots. **g** As in **b**. **h** As in **c**. **i**, **j** Strength measures for AM/CM-variants. **i** As in **a**. **j** As in **f**. **k** As in **b**. **l** As in **c**. **m**, **n** Pearson correlation of strength measures. **m** Δ Annotated (VAR/REF) for AM & AM/CM-variants (all variants which affect the annotated-donor). **n** Cryptic difference (VAR − REF) for CM & AM/CM-variants (all variants which affect the cryptic-donor).

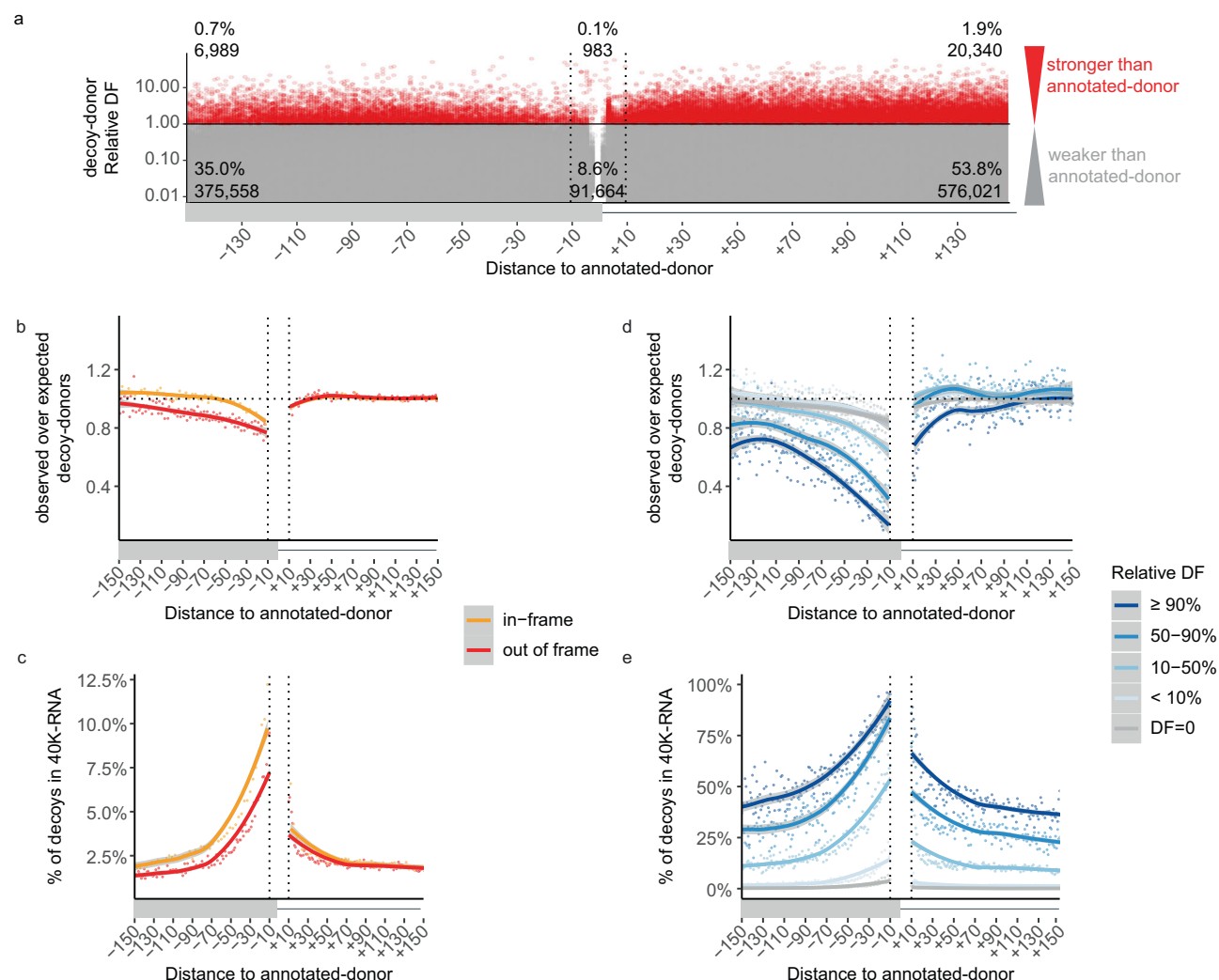

**Fig. 4 Competitive decoy-donors are specifically surrounding annotated-donors. a** The relative Donor Frequency (DF) of all decoy-donors within 150 nt of annotated-donors (decoy-donor DF / annotated-donor DF). Plots are shown for $+/-150$ nt of annotated-donors due to the steeply declining number of exons longer than this. Decoy-donors with a stronger DF score than the annotated-donor are shown in red, otherwise grey. **b** Depletion of GT decoy-splice sites (observed/expected) (see Methods and Supplementary Fig. 4). Exonic donors where use of the decoy-donor would be in-frame are shown in orange, whereas those out-of-frame (or intronic) are shown in red. GT decoy-donors show increasing exonic depletion approaching the annotated-donor, and negligible depletion in the intron. **c** Decoy-donors in-frame and closer to the annotated-donor are more likely to be present in 40,233 publicly available RNA-seq samples (40K-RNA). At each distance from the annotated splice-site, the number of decoy-donors present in 40K-RNA is divided by the total number of naturally occurring decoy-donors at that position **d** depletion of GT decoy-donors as in **b**, split according to decoy-donor DF relative to the annotated-donor (decoy-donor DF/annotated-donor DF). There is negligible depletion of decoy-donor sequences that do not exist as a bonafide donor in GRCh37 (DF = 0, grey), with increasing depletion of exonic decoy-donors closer in DF to the annotated donor (blue gradient). **e** Proportion of GT decoy-donors seen in 40K-RNA as in **c**, split as in **d**. Decoy-donors closer to the annotated-donor and with higher DF relative to the annotated-donor are more likely to be present in 40K-RNA. Lines show LOESS smoothing (locally weighted smoothing i.e., trendlines) with confidence bands in grey.

could affect measures of depletion, and/or, influence the usability of decoy-donors in this region. For example, G-repeat splicing regulatory elements (SREs) are abundant within the first ~50 nt of the intron[20–22].

We defined competitive decoy-donors as those with a DF of at least 10% that of the associated annotated-donor (see Fig. 4d, e). In the first 50 nt of the intron, competitive decoy-donors overlapping G-triplets show no depletion and conversely appear enriched (Fig. 5a, intron- orange). In contrast competitive decoy-donors not overlapping G-triplets are depleted (Fig. 5a, intron-grey). Additionally, a higher proportion of intronic decoy-donors not overlapping G-triplets are seen in 40K-RNA than those overlapping G-triplets (Fig. 5b, intron- grey). The reciprocity in these data is consistent with a masking effect of intronic G-repeat

motifs on (competitive) decoy-donors, likely due to RNA secondary structure and/or RNA binding proteins preventing their use.

Figure 5c shows an example variant in gene *GAA* (NM_000152.3:C.2646 + 2 T > A) identified in an individual affected with glycogen storage disease type II[23] that induces splicing to an exonic cryptic-donor 21 nt upstream of the annotated-donor. NNS, MES, and DF rank the decoy-donor at +57 as the strongest donor - however this donor is enveloped within a G-repeat rich region which may mask it, and accordingly is not present in 40K-RNA. SAI instead predicts use of the cryptic-donor at −21. Notably, this cryptic-donor is present in 137 samples in 40K-RNA, providing empirical evidence that despite its weak primary motif, it can be used by the spliceosome.

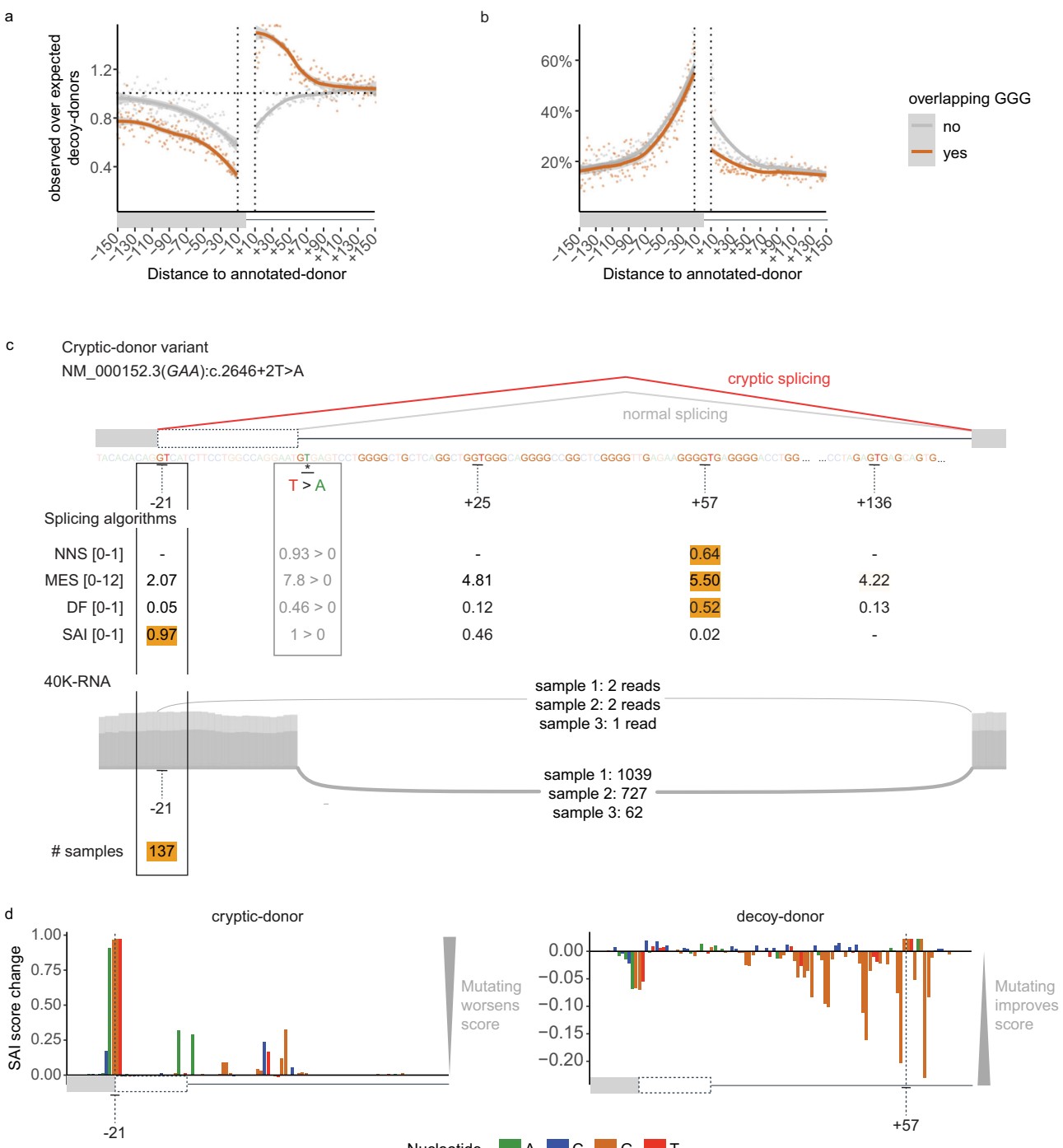

**Fig. 5 Utility of 40K-RNA to identify decoy-donors able to be used by the spliceosome. a**, **b** Depletion (**a**) and proportion in 40K-RNA (**b**) of GT decoy-donors that do or do not overlap G-triplets. Calculated as in Fig. 4b, c. Decoy-donors overlapping G-triplets are depleted in the exon but not in the intron, where they show enrichment because: 1) G repeats are enriched in the first 50 nt of the intron (see supplementary Fig. 3) and 2) donor sequences are commonly G-rich. Plots are limited to decoys with relative DF > 0.1 (defined as competitive with the annotated-donor, see Fig. 4d, e). Lines show LOESS smoothing (locally weighted smoothing i.e. trendlines) with confidence bands in grey. **c** Top: Schematic of an AM-variant identified in an individual with glycogen storage disease type II associated with gene *GAA* (NM_000152.3:C.2646+2 T > A)[23] with algorithm scores for annotated- (REF > VAR) decoy- and cryptic-donors. NNS NNSplice, MES MaxEntScan, DF Donor Frequency, SAI SpliceAI. The strongest donor for each algorithm (score range shown in square brackets) is coloured orange. Below: Sashimi plot from three GTEx RNA-seq samples identifying use of the -21 cryptic-donor as present in 40K-RNA (At least 1 read is detected in 137/30753 samples with detectable expression of transcript). SpliceAI (SAI) correctly scores the −21 cryptic-donor as the most likely cryptic-donor. **d** Result of SAI in silico mutagenesis showing the bases contributing to predicted strength of the −21 cryptic-donor (left) and +57 decoy-donor (right). SAI score change denotes the decrease (if positive) or increase (if negative) on the predicted strength of the donor when that nucleotide is mutated (see methods). Note that the presence of the cryptic-donor at −21 and intronic G-repeats negatively impact the score of the +57 decoy-donor according to SAI.

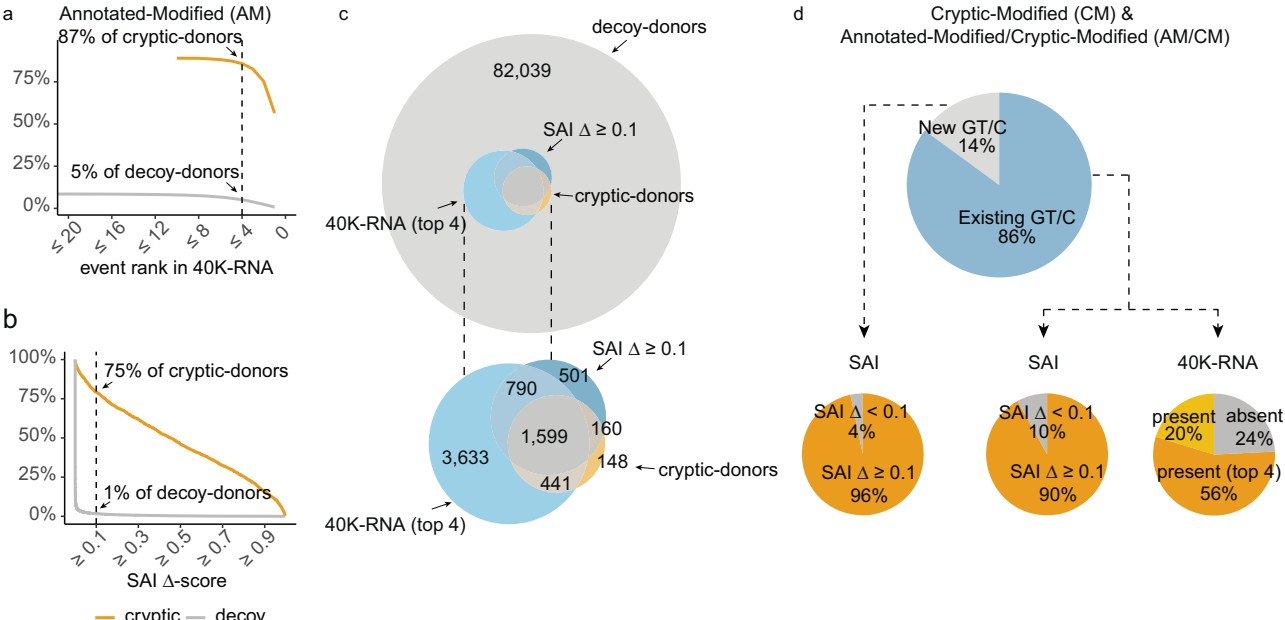

**Fig. 6 40K-RNA potently informs cryptic-donor activation. a, b** The percentage of Authentic-Modified (AM) cryptic-donors correctly predicted, and decoy-donors incorrectly predicted at different cut-offs for (**a**) event rank in 40K-RNA and (**b**) SpliceAI (SAI) Δ score (donor$_{VAR}$ -donor$_{REF}$). **a** The dotted line denotes cryptic- and decoy-donors predicted using a cut-off of events ranked 4 or less. 87% of cryptic-donors activated by AM-variants are among the top 4 events present in 40K-RNA within 250 nt of the annotated-donor, with 95% of decoy-donors not in the top 4 events. **b** While SpliceAI (SAI) outperforms other algorithms using a cut-off of Δ scores 0.1 and above (see supplementary Fig. 6a), it predicts only 75% of cryptic-donors (SAI Δ ≥ 0.1). SAI accurately excludes 99% of decoy-donors. **c** (top) Venn diagram showing the overlap of cryptic-donors with those predicted by SAI or our 40K-RNA method, among the entire pool of decoy-donors. (bottom) magnification of internal Venn. **d** Cryptic-donors activated by CM- and AM/CM-variants. (top) New GT/C (Light grey) denotes variants creating a GT or GC essential splice-site motif (40K-RNA is inherently unsuitable for these variants). Existing GT/ C (blue) denotes variants that modify a decoy-donor with a pre-existing GC or GT essential splice-site. (bottom) For 40K-RNA, the orange segment denotes cryptic-donors in the top 4 events, yellow segment denotes where the cryptic-donor is present in 40k-RNA but is not in the top 4 events, and grey denotes cryptic-donors absent from 40K-RNA.

SAI in silico mutagenesis of the cryptic-donor at -21 and decoy-donor at +57 show that SAI deep-learning perceives the negative impact of the G-repeats on the usability of the +57 decoy-donor (Fig. 5d). Intronic G-repeats are known examples of SREs[20,24] (see Fig. 5 and additional examples supplementary Fig. 5c–f). Whether or not a cryptic donor can be used is influenced by a constellation of features: the consensus donor sequence, as well as proximal and more distal splicing regulatory elements. Regulatory elements are not factored by many algorithms, though may be identified by SAI, likely underpinning its enhanced capabilities in recognition of usable (cryptic) splice-sites. In contrast, 40K-RNA uses empirical evidence from RNA-Seq data that reveals which cryptic splice-sites are usable in the context of the specimens tested.

**90% of cryptic-donors in AM-variants are present in 40K-RNA.** We assessed whether 40K-RNA provides a viable means to prioritise cryptic-donors likely to be activated in the context of a genetic variant affecting the annotated-donor (i.e. AM-variants). 90% of AM-variant activated cryptic-donors are present in 40K-RNA, while 91% of unused decoy-donors are absent. Therefore, 40K-RNA provides potent predictive information with respect to both true positives (cryptic-donors) and true negatives (decoy-donors). Notably, while cryptic-donors were observed in multiple independent samples across 40K-RNA, they were typically very low frequency splice-junctions (44% had a maximum of 4 reads or less in any one sample, supplementary Fig. 6b).

We chose the top 4 40K-RNA events at each splice-junction (or all events if there were less than 4 detected) as our predicted

cryptic-donors as this maximised sensitivity (87%) without compromising specificity (95%) (Fig. 6a). Use of 40K-RNA had a higher sensitivity than all four algorithms assessed (Fig. 6a, b, supplementary Fig. 6a). The sensitivity of 40K-RNA is inherently influenced by read-depth of the target transcript: more than 85% of cryptic-donors are detected in transcripts with a read depth of >250 for the annotated exon-exon splice-junction (normal splicing); whereas only 29% of cryptic-donors are detected in 40K-RNA in transcripts where normal splicing had a maximum read count of <100 (supplementary Fig. 6c). Consequently, we assessed SAI as a complementary approach for situations where our empirical method is underpowered or not well suited.

We define SAI prediction of cryptic-donor activation as a donor-gain Δ−score of 0.1 or greater, which accurately predicts 75% of cryptic-donors and inaccurately predicts only 1% of decoy-donors (Fig. 6b). SAI showed higher sensitivity then NNS, and comparable sensitivity to MES and DF, while greatly improving on their specificity (supplementary Fig. 6a). However, the sensitivity of SAI is compromised for cryptics at increasing distance from the annotated-donor - only 55% of cryptic-donors further than 100 nt from the annotated splice site had a Δ-score above 0.1 (supplementary Fig. 6d, e). If we take the union of SAI and 40K-RNA cryptic-donor predictions (i.e., cryptics predicted by either of the two methods), we accurately predict 2210/2389 (93%) of cryptic-donors (Fig. 6c) and inaccurately predict 6% of unused decoy-donors.

Use of 40K-RNA has caveats for CM-variants and AM/CM-variants, and cannot be used for variants that create a GT (or GC) motif. However, for the subset of CM-variants and AM/CM-variants where the variant modifies the extended splice site region

of an extant GT/C decoy-donor (1525 variants, Fig. 6d, top- blue), 76% are present in 40K-RNA, with 56% in the top 4 events.

40K-RNA is least sensitive for variants that most significantly impact the strength of the cryptic-donor: For $D^{+2}$ CM-variants, only 32% of the cryptic-donors are present in the top 4 events, as compared to 85% for $E^{-3}$ variants (supplementary Fig. 6f; $E^{-3}$ is the third to last exonic base). Accordingly, even if a GC decoy-donor is not present in 40K-RNA, conversion by a variant to a GT donor presents high risk for cryptic-activation. SAI performed well for CM-variants and AM/CM-variants, correctly predicting 96% of variants that created an essential donor motif and 90% which modified an existing essential motif (Fig. 6d).

## Discussion

The ultimate goal of splicing predictions is to determine if and how a genetic variant will induce mis-splicing of pre-mRNA. Even for essential splice-site variants that almost invariably cause mis-splicing, consideration of probable consequences of the variant is critical for pathology application of the ACMG-AMP code PVS1[25] (null variant due to presumed mis-splicing of the pre-mRNA) and of equal importance to strategise functional testing for RNA diagnostics[14]. While activation of a cryptic-donor 6 nucleotides away will remove or insert two amino-acids, activation of a cryptic-donor 4 nucleotides away will induce a frameshift, with vastly different implications for pathology interpretation.

We learned five key lessons from our analyses of 4811 cryptic-donor variants in 3399 genes: (1) Decoy-donors that show evidence of natural stochastic use by the spliceosome in population-based RNA-Seq data (i.e., are present in 40K-RNA) have the greatest probability of activation as cryptic-donors. (2) Decoy-donors closer to the annotated splice site are inherently more likely to be used by the spliceosome, likely due to the presence of all required sequence features that are facilitating use of the annotated donor.

(3) Cryptic-donors do not necessarily need to be stronger than the annotated-donor to present substantive risk for mis-splicing, with decoy-donors only 10% of the strength of the annotated-donor able to compete for spliceosomal binding. (4) Intronic G-repeats can diminish the likelihood of spliceosomal recognition and use of intronic decoy splice sites. (5) SAI's deep-learning appreciates the broader sequence context determining spliceosomal usability of a cryptic-donor, though less accurately predicts activation of more distal cryptic-donors (>100 nt from the annotated-donor).

SAI's deep learning presents a major improvement in predicting cryptic-donor activation. However, use of SAI in a pathology context is limited by the challenge of deriving a clinically-meaningful interpretation of a number on a 0–1 scale, without independently verifiable evidence. In contrast, 40K-RNA provides an accurate, evidence-based means to rank cryptic-donors likely to be activated by genetic variants.

Brandão et al.[26] used deep sequencing of twelve major cancer susceptibility genes to catalogue all alternative and aberrantly spliced transcripts. They found variant-activated cryptic splicing was often seen at much lower levels in disease controls, suggesting that the dominant transcript in rare disease may be seen as a stochastic mis-splicing event in other samples. We use this insight, mining the breadth of publicly available RNA-seq data across numerous tissues to comprehensively catalogue stochastic cryptic splicing events across all genes.

The heightened sensitivity and empirical nature of using 40K-RNA is of vital importance for pathology assessment of variants affecting the essential donor splice-site, as not considering a likely cryptic-donor activated can lead to profound complications in variant interpretation. Prospectively, the sensitivity of 40K-RNA can be enhanced by ultra-deep sequencing. It is also easy to envisage extending the method to predict other mis-splicing events such as exon skipping, and mis-splicing events at the acceptor splice site. 40K-RNA can reliably identify distal cryptic-donors with high likelihood of activation, which may not be identified by SAI. Conversely, SAI can reliably identify cryptic donors with high likelihood of activation not detected in 40K-RNA, due to low read depth of the target gene.

In conclusion, we define an accurate, evidence-based method to predict cryptic-donor activation in the context of a variant affecting the annotated-donor, based on stochastic mis-splicing events observed in 40,233 publicly available RNA-seq samples (40K-RNA). We provide a web resource that reports and ranks the most commonly (mis)used cryptic donors proximal to every ensembl annotated-donor[27] (https://kidsneuro.shinyapps.io/splicevault-40k/). Our research establishes that for AM-variants, if a cryptic-donor is activated, in 87% of cases it will be among the top 4 events. We hope this evidence-based method may improve clinical interpretability of donor variants.

## Methods

**Creating a database of cryptic-donor variants.** Variants were derived from several sources: (1) 439 variants curated from literature, predominantly comprised of 364 variants in DBASS5[15] and supplemented by curation from published literature of 75 additional variants[28,29] (2) 4372 variants derived from RNA-seq studies: Variant-associated aberrant cryptic-donor activation detected from RNA-seq data identified by SAVnet in somatic tumor samples ($n = 3259$)[16] and 1113 variants identified in GTEx samples by spliceAI and verified using RNA-seq data[11]. The following inclusion criteria applied: (1) Variants had to occur within $E^{-4}$-$D^{+8}$ of the annotated or the cryptic-donor, otherwise they were excluded as outside the bounds of this analysis. (2) annotated cryptic-donors were within the same exon/intron as the variant (i.e., between the 5′ end of the exon and 3′ end of the intron surrounding the affected donor). (3) The annotated cryptic-donor VAR sequence had to have an essential GT/GC dinucleotide at $D^{+1}/D^{+2}$, to minimise mis-annotated variants being included.

*Annotating variant categories.* We annotated variants with categories we defined– if the variant occurred within $E^{-4}$-$D^{+8}$ of the annotated-donor, it was an AM-variant, if it occurred within $E^{-4}$-$D^{+8}$ of the cryptic-donor it was a CM-variant, and if it occurred within $E^{-4}$-$D^{+8}$ of both the annotated- and cryptic-donor it was an AM/CM-variant. For 37/373 of AM/CM-variants, an additional unmodified cryptic-donor was activated, in addition to the cryptic-donor modified by the variant- these were excluded from analyses.

*Compiling annotated-, cryptic- & decoy-donor sequences.* The R package BSgenome.Hsapiens.1000genomes.hs37d5[30] was used to extract (up to) 500 nt of genomic sequence preceding and succeeding the annotated-donor (GRCh37). For each variant in the cryptic-donor database, we extracted up to 250 exonic nucleotides in the 5′ direction (i.e., if the exon was only 50 nt the window of analysis would be 50 nucleotides), and up to 250 intronic nucleotides in the 3′ direction, in the same fashion (Fig. 1a).

From the (up to) 500 nt of sequence we pulled $E^{-4}$-$D^{+8}$ sequences for the annotated- and cryptic-donor before and after each variant (REF and VAR respectively). We also identified any other essential donor dinucleotides (i.e., GT or GC) which were present in the sequence and extracted the $E^{-4}$-$D^{+8}$ sequence surrounding them. These sequences we define as decoy-donor- sequences containing the essential donor dinucleotides (i.e., a GT or a GC) but which weren't utilised by the spliceosome as a result of the variant (Fig. 1a, lower). For intronic decoy-donors, we excluded any which would result in an intron too short to be spliced (as defined by the 1st percentile for intron length in the human genome = 80 nt)[31]. Importantly, without additional filtering, no cryptic-donors in the database violated this rule.

**Algorithms for splice site strength.** We retrieved predicted scores for annotated-donors, cryptic-donors and decoy-donors in the database in both the REF and VAR sequence context, for four algorithms. (1) MaxEntScan (MES)[13] scores were retrieved using the perl script provided at http://hollywood.mit.edu/burgelab/maxent/Xmaxentscan_scoreseq.html. MES scores below 0 were standardised to 0 as predicted non-functional splice sites (2) NNSplice (NNS)[12] scores were retrieved using the online portal (https://www.fruitfly.org/seq_tools/splice.html), set to human, with default settings (i.e., a minimum score of 0.4, with any scores below predicting a non-functional splice site) (3) SpliceAI (SAI)[11] scores were retrieved using a script adapted from the SAI GitHub repository (https://github.com/Illumina/SpliceAI) which allows spliceAI to score custom sequences. We rounded

the scores to three decimal places, and scores at 0 when rounded as such (i.-e., < 0.01) were termed non-functional splice site predictions. (4) Donor Frequency (DF) calculates the median frequency among four 9 nt windows of sequence spanning the donor (see supplementary Fig. 1b, c) converted to a cumulative percentile distribution scale. DF measures donor strength based on how many annotated-donors in the human genome have the exact same sequence. An example of a DF calculation is shown in supplementary Fig. 1c, where a median DF raw value of 179 lies at the 31st percentile of a cumulative frequency distribution. After assessing several window lengths (supplementary Fig. 1a) we used 9nt windows as optimally encompassing the splice site.

**Naturally occurring decoy-donors**. Our set of naturally occurring human decoy-donors were derived from the set of all canonical Ensembl transcripts (Release 75)[27], with first and last introns and single exon transcripts removed. We filtered the set so that junctions were within the open reading frame for the given gene, so we knew that cryptic splicing here would affect the protein. We also removed exons with alternative 5′ or 3′ ends already annotated in different transcripts. We used these criteria to form a set of 142,014 canonical exon-intron junctions that are not alternatively spliced (or at least not annotated as such). We extracted sequences surrounding annotated-donors and extracted all decoy-donors just as for the cryptic database (see methods section creating a database of cryptic-donor variants).

**Decoy-donor depletion**. Decoy-donor depletion was calculated using a method we adapted from a previous study[17] that controls for dinucleotide frequencies (supplementary Fig. 4). For exonic sequences, we took up to 150 nt or the maximum length of the exon, whichever was shorter (and similarly for the intron, stopping 50nt from the acceptor). We limited analysis to 150nt of exonic sequence as the majority of exons are shorter than this. We then shuffled exonic and intronic sequences separately, maintaining dinucleotide frequencies (using shuffle_sequences with euler method from the universal motif R package[32]). We performed the shuffle 15 times and took the average count of decoy-donors at each nucleotide position as our expected count at this position. The observed count of decoy-donors was then divided by the expected count at each position.

**Creating 40K-RNA**. We had two sources of data for 40K-RNA- RNA-seq data from Intropolis[19] and GTEx[18]. Intropolis is a set of ~42 M splice-junctions found across 21,504 human RNA-seq samples from the Sequence Read Archive (SRA). Samples were aligned using Nellore et al. annotation-agnostic aligner Rail-RNA[33]. Intropolis was downloaded from its dedicated github repository (https://github.com/nellore/intropolis). Per sample splice-junction files were obtained from GTEx (phs000424.v8.p2 [https://www.ncbi.nlm.nih.gov/projects/gap/cgi-bin/study.cgi?study_id=phs000424.v8.p2]). Using Datamash[34], splice-junction read counts were summarised across all samples for each unique splice-junction and translated from GRCh38 to GRCh37 using liftOver[35].

For each set of splice-junctions (Intropolis and GTEx), we cross-referenced and located junctions within ensembl transcripts. We filtered to cryptic-donor events by scanning for any unannotated donors used between the 5′ end of the exon and the 3′ end of the intron for that respective exon-intron junction, where the junction also spliced to the next annotated acceptor. Events from the two sources were merged, sample counts were tallied across the two datasets, and splice-junctions present in at least 3 samples and representing cryptic-donor use within 250 nt of any annotated-donor were retained.

**Sashimi plots**. For Fig. 5c, and S6b, c sashimi plots were generated using 3 GTEx bam files for each example, each from the tissue with the highest TPM for the respective gene. Sashimi plots were created using ggsashimi[36].

**SpliceAI in silico mutagenesis plots**. For Fig. 5d and S6b, c we performed the in silico mutagenesis method described by Jaganathan et al[11]. That is, the importance score of each nucleotide was calculated as:

$$s_{actual} - \frac{s_A + s_C + s_G + s_T}{4} \tag{1}$$

where $s_{actual}$ is the score calculated on the genuine sequence, and $s_A$, for example, is the score calculated when an A is substituted at this position.

**Reporting summary**. Further information on research design is available in the Nature Research Reporting Summary linked to this article.

## Data availability

The variants used in the cryptic-donor database are provided in the Source Data file. 40K-RNA is available as a web-resource at: https://kidsneuro.shinyapps.io/splicevault-40k/. Additionally, the full dataset is available under restricted access to limit hosting costs. Access can be obtained by creating a google cloud billing account and downloading at this link using google cloud tools- https://storage.googleapis.com/misspl-db-data/misspl_events_40k_hg19.sql.gz. The GTEx v8 data used in this study were obtained from

dbGaP accession number phs000424.v8.p2 [https://www.ncbi.nlm.nih.gov/projects/gap/cgi-bin/study.cgi?study_id=phs000424.v8.p2]. Intropolis data used in this study were obtained from the dedicated GitHub repository https://github.com/nellore/intropolis. Source data are provided with this paper.

## Code availability

All code required to replicate figures in the study are available in a GitHub repository: https://github.com/kidsneuro-lab/cryptic_donor_prediction. Additionally, code required to create 40K-RNA is available in a separate repository https://github.com/kidsneuro-lab/40K-RNA.

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

## Acknowledgements

This project was supported by a National Health and Medical Research Council of Australia Senior Research Fellowship (S.T.C. APP1136197) and Ideas Grant (S.T.C. APP1186084). R.D. is supported by a University of Sydney Research Training Program Scholarship and Merit Award Supplementary Scholarship. The Genotype-Tissue Expression (GTEx) Project was supported by the Common Fund of the Office of the Director of the National Institutes of Health, and by NCI, NHGRI, NHLBI, NIDA, NIMH and NINDS.

## Author contributions

Data curation and analysis: R.D. and H.J. Funding acquisition and supervision: S.T.C. Visualization: R.D. Writing – original draft: R.D. Writing review and editing: R.D. and S.T.C.

## Competing interests

S.T.C. and H.J. are named inventors of Intellectual Property (IP) described in part within this manuscript owned jointly by the University of Sydney and Sydney Children's Hospitals Network. S.T.C. is director of Frontier Genomics Pty Ltd (Australia) who have licenced this IP. S.T.C. receives no payment or other financial incentives for services provided to Frontier Genomics Pty Ltd (Australia). Frontier Genomics Pty Ltd (Australia) has no existing financial relationships that will benefit from publication of these data. The remaining co-authors declare no conflicts of interest.
