## [Peer Review File · Nature Communications]

Empirical prediction of variant-activated cryptic splice donors using population-based RNA-Seq dataReviewers' Comments:

Reviewer #1:

Remarks to the Author:

Using large-scale RNA-seq splice-junction data sets, the authors proposed an empirical approach to identify features related to variant-associated cryptic donor activation. They observed that decoy-donors are more likely to be activated as cryptic-donors if they are splice competent. Additionally, they reported that proximity to authentic splice site increases the likelihood of a decoy-donor functioning as a cryptic-donor. Using this additional knowledge, the authors reported a substantial improvement of identifying cryptic donors using a combined approach of splice competent and SpliceAI. Compared to currently available methods, their developed method is able to identify variant-associated cryptic donor activation with 87% sensitivity and 95% specificity.

The manuscript is well-written and the topic should be of interest to a wide-range of audience. The study showcased the importance of utilizing RNA-seq data to better understand the mechanisms of alternative splicing. The additional knowledge gained from RNA-seq data can greatly complement machine learning/deep learning based models. However, I have some minor comments regarding the clarity of the paper.

1. How did you determine the E-4 to D+8 window to study? Since based on Figure 1.e, it seems for AM/CM variants, there is an increasing trend of variant count when moving to the 5' direction of the exon.
2. It is not clear to me why you used a 150 nt window to calculate decoy donor depletion whereas you used 250 nt window for your other analyses.
3. As extensively used in the manuscript, the 'strong' donor needs more clear definition.
4. Fig. 2 a&b, I am confused by the presence of cryptic donors that have 0 distance from the authentic-donor. Are there overlaps between the cryptic and authentic donor sites?
5. A brief highlight and conclusion in the end to inform readers the potential applications of the proposed method/dataset should be helpful.

Reviewer #2:

Remarks to the Author:

In this manuscript the authors take a careful look at how genetic variants can cause the use of cryptic splice donors (SD). They do this by collating both large-scale RNA-seq studies and the literature. They distinguish variants modifying the "authentic" SD (AM) from those that modify the cryptic SD (CM), as well as those modifying both. They characterize various interesting properties of these variants: their position relative to the affected SD, the relative positions of the authentic and cryptic SD and so on. They consider the ability of 3 algorithms (MaxEntScan, NNSplice and SpliceAI) to predict the variant effects via their scoring of the SDs. They additionally define a score ("DF") based on whether that cryptic SD is ever observed in >40k uniformly processed RNA-seq samples. This metric is particularly useful for AM variants and still shows some utility for CM (although less as expected since the specific CM variants are unlikely to be represented in the RNA-seq data).

This paper was well written and clear, with an appropriate level of detail split between the methods and results. The figures are also excellent. I have only minor comments:

- Have you seen "authentic" used elsewhere? I am used to calling these "canonical" SD. I don't hate "authentic" but I would prefer papers stick to standard nomenclature when possible.
- "with deletions ranging from 1 to 57 nts in length" make it clearer this means deletions in the transcript not indels in the DNA (as I originally thought reading this sentence!)
- REF and VAR should be REF and ALT since this is more standard
- you should describe HOW you combine DF and SAI.
- "We used the 1000genomes Phase2 Reference Genome Sequence (hs37d5) version of hg1927 implemented in RStudio" I would call this "in R"... and you implemented the genome?!
- "as defined by the 5th percentile for intron length in the human genome"... which is?

- "Decoy-donor depletion was calculated using an adapted method" adaptive?
- aligned Rail-RNA -> aligner
- "translated from GRCh38 to GRCh37" urgh come on it's 2021. But I realize it would be a lot of work to redo this.

The work seems technically solid to me, if not extremely innovative. The observation that large RNA-seq cohorts are valuable for this task is valuable and insightful however.

- "Counter-intuitively, less than half of activated cryptic..." This is only counter-intuitive to me if the canonical SD is now not used at all in favor of the cryptic SD, but I don't think that is the case typically here? If the cryptic SD is close in strength to the canonical it is intuitive to me the cryptic will be chosen some proportion of the time. Splicing is somewhat noisy after all.
- "85-99% weaker" There is an awkward question of scale here. What does splice site strength mean? I'm not sure there's a great answer since it will depend on the specific algorithm. e.g. for spliceAI it means "how probable is this to be an annotated SD", but MES takes a more enrichment based approach. Equivalently one could think of considering the log odds instead of the probability... 90% would then mean something different. I don't think there is a good answer (no fault of the authors) but it might be worth discussing a bit.
- "NNS, MES, and DF rank the decoy-donor at +57 as the most" ... it is not surprising to me that NNS and MES get this wrong since they don't know (or even see) the G-rich SRE. But it is surprising to me that DF gets this wrong: shouldn't +57 not be used in the RNA-seq database following your logic?

Discussion:

- "determining spliceosomal 'usability' of a decoy splice-site, including 2-4 above,". I agree SpliceAI models your points 3 and 4, but not 2. When it makes a prediction it doesn't know anything about where the canonical SD is.

I recommend acceptance on the condition the authors make their literature curated set of known (variant, canonical SD, cryptic SD) publicly available, since this will be a nice resource for the community. Oh a final point: I don't like the title. Donor -> Splice donor, and replace the numbers with just "high". The precise numbers are too dependent on many factors and not particularly meaningful in themselves.

Reviewer #3:

Remarks to the Author:

The manuscript by Dawes et al presents an interesting overview of variant-associated cryptic splicing donors by integrating the analysis of 4811 genetic variants with available RNA-seq data. Although some interesting results are presented, I have some serious concerns as detailed below.

1) The title is misleading, the approach proposed by the authors is not general and can not be applied for the "prediction of variant-associated cryptic-donors" as it is. It is my understanding that RNA-sequencing data are required, which are not always available in clinical studies. The authors claim that their method can be used to evaluate and understand the pathological potential of variants currently considered to be VUS, however this is not the case, since the approach they propose can be used only for a limited selection of cases.

2) The evaluation of the sensitivity and specificity of the approach proposed by the authors is not carried out in a rigorous manner. No cross fold validation, training and/or testing sets are used to provide an unbiased estimate. No independent dataset is used (for example one based on long read sequencing technologies). No wet-lab validation is performed. With the paper in the current form authors can not claim that they have 87% sensitivity and 95% specificity.

3) The method for the prediction of cryptic donor sites is based on the integration of a pre-existing

tool SpliceAI with RNA-sequencing data. Per se it does not provide the required level of scientific advancement/novelty required for a top-level scientific publication. Additionally levels of sensitivity depend dramatically on depth of sequencing of the RNAseq. Finally, it is the opinion of this reviewer that availability of RNA sequencing data is not a common feature in several clinical studies. Additionally it not clear why the authors decided to use SpliceAI and not any other tool/tools that they considered in the study.

4) Most of the analyses are purely descriptive (as in they present descriptive statistics about splicing donor sites), the novel approach proposed by the authors Donor Frequency (DF), simply captures the nucleotide composition around splice donor sites and is not substantially different from any consensus-PWSM (positional weight scoring matrix) based method. As such the analyses based on DF score do not provide interesting additional insights apart from the fact that a specific consensus sequence is preferentially observed.

5) Splicing is also mediated/controlled by epigenetic factors and/or regulation, so it would be appropriate do discuss the approach proposed by the user should be at least discussed in this context

Minor remarks

Some references are incomplete (see ref. 12 or 14, where the journal is missing). Please check all.

REVIEWER COMMENTS

As a general note, in the writing of our paper we had a difficult job trying to devise easily understandable terminology for several concepts. We can see from our reviewer's comments a need for further improvement. Therefore, we have refined our language throughout the manuscript, for example:

- substituting 'annotated' for 'authentic' donors
- defining our database of splice-junctions derived from 40,233 RNA-seq samples as '40K-RNA' and removing use of the term 'splice competent'.
- We take care to define each new term in the paper with (what we hope is) improved clarity, and ensure we use the same terminology consistently throughout.

We now also include a glossary in the methods section which we think may be helpful for our readers to explain a number of new terms:

PG18, Line 1010

Glossary

Decoy-donor: Any essential donor dinucleotide (GT/GC) that is not an annotated-donor.

Cryptic-donor: A decoy-donor shown to be activated (i.e. used by the spliceosome) by a genetic variant.

Annotated-Modified (AM): A genetic variant modifies the annotated-donor resulting in activation one or more unmodified cryptic-donors.

Cryptic-Modified (CM): A genetic variant modifies a cryptic-donor and does not affect the annotated-donor.

Annotated-Modified/Cryptic-Modified (AM/CM): A genetic variant that simultaneously modifies the annotated-donor and nearby cryptic-donor.

Donor Frequency (DF): A measure of donor strength based on how many annotated-donors in the human genome have the exact same sequence.

Competitive decoy-donor: A decoy-donor with a DF score at least 10% the score of the nearby annotated-donor.

40K-RNA: An aggregated database of splice-junctions detected across 40,233 publicly available RNA-seq samples.

Reviewer #1 (Remarks to the Author):

R1.1 *How did you determine the E-4 to D+8 window to study? Since based on Figure 1.e, it seems for AM/CM variants, there is an increasing trend of variant count when moving to the 5' direction of the exon.*

Our selection of E-4 to D+8 region is a fundamental element of our analysis, and we agree that we do not articulate clearly enough how and why it was selected. We have reworded to improve clarity:

PG3, Line 55:

“We define the extended donor splice-site region as spanning the fourth to last nucleotide of the exon (E^{-4} , E = exon) to the eighth nucleotide of the intron (D^{+8} ; D = donor), as constraint on sequence diversity eases beyond this point (Supplementary Fig. 1).”

PG28, Line 1315, Figure S1 legend:

*“Supplementary Fig. 1 Calculation of Donor Frequency as a measure of donor strength. a-b) Frequency of unique combinations of donor sequences at each position of the exon-intron junction, spanning 6, 7, 8, 10 (a) or 9 (b) consecutive nucleotides. Black bars denote windows overlapping the $E-4$ - $D+8$ donor sequence window. Four sliding windows of 9 nt spanning the **annotated-donor** (coloured black), spanning 12 nt from the fourth-to-last exonic base ($E-4$; E = exon) to the eighth intronic base ($D+8$; D = donor), were used for DF calculation. **These windows were chosen due to the jump in sequence diversity seen with windows upstream of/including $E-5$, and downstream of/including $E+9$.** d) Donor Frequency is calculated as the median frequency across each 9nt window, converted to a cumulative percentile distribution. **DF measures donor strength by how many annotated-donors in the human genome have the exact same sequence. In this example, a median DF raw value of 179 lies at the 31st percentile of a cumulative frequency distribution.**”*

To clarify here:

- a) Figure S1 shows evolutionary constraint upon sequence combinations across the donor splice-site varying window length from 6 nt to 10 nt. Constraint upon sequence combinations reduces sharply at the peripheries (most obvious with short 6 nt or 7 nt windows). The jump in sequence diversity seen with windows upstream of/including $E-5$, and downstream of/including $E+9$, defined the boundaries of $E-4$ to $D+8$ as the most constrained region.
- b) Figure 1Ci and 1Di both show sharply decreasing numbers of variants towards the periphery of our window, proportional to the sequence constraint of these bases, and providing further justification for using the $E-4$ to $D+8$ window. Figure 1Ei is the exception, however the higher numbers of splice-altering $E-4$ and $E-3$ variants is due specifically to the mechanism highlighted in Figure 2C, whereby the +5 and +6 of the annotated-donor now becomes the +1 and +2 GT of the cryptic-donor.

R1.2 *It is not clear to me why you used a 150 nt window to calculate decoy donor depletion whereas you used 250 nt window for your other analyses.*

As the 50th percentile of human exon lengths is 123 nt, the number of exons declines exponentially with longer exon lengths (as can be seen in figure 2D). We originally plotted +/- 250 nt. Decoy depletion on the exonic side also did not change between 150 and 250 nt, though the plots got ‘messier’, due to declining numbers of exons to input into the observed / expected calculation. Additionally, decoy-depletion on the intronic side was uniform (i.e. a straight line). Thus, for simplicity and to communicate the most relevant findings, we

present a 150 nt window. To address this point, we emend the legend for Figure 4 to clarify our reasoning for presenting a 150 window rather than 250 nt.

PG 11, Line 400, Figure 4 legend:

*“The relative DF of all decoy-donors within 150 nt of **annotated**-donors (decoy-donor DF / **annotated**-donor DF). Plots are shown for +/-150 nt of **annotated**-donors due to the steeply declining number of exons longer than this”*

R1.3 *As extensively used in the manuscript; the ‘strong’ donor needs more clear definition.*

The term ‘strong’/ ‘stronger’ is always in the context of comparing measures of splice-site strength by the *same algorithm*. We initially used the term ‘scored as stronger by the respective algorithm’ in drafting the manuscript, though felt this was a little unwieldy. Therefore, we address this point by providing a specific definition at first mention:

PG 8, Line 240:

“The four algorithms use different methods to measure the intrinsic ‘strength’ of a given donor splice-site. In the following discussion we use the term ‘stronger’ and ‘weaker’ to denote a donor that has a higher or weaker score, respectively, according to that algorithm. Comparisons such as ‘weaker by >50%’ denote that the donor score has been reduced by more than half by the variant”

R1.4 *Fig. 2 a&b, I am confused by the presence of cryptic donors that have 0 distance from the annotated-donor. Are there overlaps between the cryptic and annotated donor sites?*

We acknowledge the confusion, due to each bar in these histograms corresponding to a 10 nt window. Thus, cryptic donors positioned between +/- 5 nt of the annotated-donor appeared in the ‘0’ tick label. To improve clarity, we have split histograms 2a, b (and d) into intronic and exonic cryptics so that the bars do not cross over ‘0’.

R1.5 *A brief highlight and conclusion in the end to inform readers the potential applications of the proposed method/dataset should be helpful.*

We’ve added a concluding paragraph to the end of the discussion (line 454) to address this point, which was also raised by R3:

PG 18, Line 1002

“In conclusion, we define an accurate, evidence-based method to predict cryptic-donor activation in the context of a variant affecting the annotated-donor, based on stochastic mis-splicing events observed in 40,233 publicly available RNA-seq samples (40K-RNA). We provide a web resource that reports and ranks the most commonly (mis)used cryptic donors proximal to every ensembl annotated-donor²⁰ (link). Our research establishes that for AM-variants, if a cryptic-donor is activated, in 87% of cases it will be among the top 4 events. We hope this evidence-based method may improve clinical interpretability of donor variants.”

(Link will be provided upon acceptance of the manuscript)

Reviewer #2 (Remarks to the Author):

R2.1 *Have you seen "authentic" used elsewhere? I am used to calling these "canonical" SD. I don't hate "authentic" but I would prefer papers stick to standard nomenclature when possible.*

We used the terminology “authentic” rather than “canonical” as a more general term for any annotated-donor, as variants in our database didn’t necessarily affect canonical transcripts, and our 40K-RNA similarly includes non-canonical as well as canonical transcripts. In consideration of our reviewer comments, we substitute the term “authentic” with “annotated”. We feel this nomenclature is more explicit and intuitive and is now used throughout the paper.

R2.2 *"with deletions ranging from 1 to 57 nts in length" make it clearer this means deletions in the transcript not indels in the DNA (as I originally thought reading this sentence!)*

We apologise for the lack of clarity. This sentence does in fact refer to deletions in the DNA and we have now modified the text within the results section to clarify.

PG 4, Line 82

*“AM-variants **which are SNVs and DNA insertions** commonly affect the E^1 , D^{+1} , D^{+2} and D^{+5} positions of the annotated-donor (Figure 1Ci), **and AM-variants which are DNA deletions** ranged from 1 to 57 nts in length (Figure 1Cii)...”*

PG 4, Line 92

*“AM/CM-variants **which are SNVs and DNA insertions** also most frequently affect the D^{+2} position (122/373) of the cryptic-donor... **AM/CM-variants which are DNA deletions** range from 1 to 36 nts in length (Figure 1Eiii)...”*

R2.3 *REF and VAR should be REF and ALT since this is more standard*

While alternate (ALT) is more standard terminology in data science, variant (VAR) is the standard clinical terminology in reference to DNA variants identified by genomic sequencing. Therefore, we prefer to maintain use of the term VAR in order to reach our clinical genetics readership, in the hope of maximising the clinical applicability of our study.

R2.4 *you should describe HOW you combine DF and SAI.*

Agreed. We have clarified in the text:

PG 16, Line 879

"If we take the union of SAI and 40K-RNA cryptic-donor predictions (i.e., cryptics predicted by either of the two methods), we accurately predict 2210/2389 (93%) of cryptic-donors (Figure 6Aiii) and inaccurately predict 6% of unused decoy-donors."

R2.5 *"We used the 1000genomes Phase2 Reference Genome Sequence (hs37d5) version of hg1927 implemented in RStudio" I would call this "in R"... and you implemented the genome?!*

We agree and have revised this (clumsy) sentence:

PG 20, Line 1070

"The R package BSgenome.Hsapiens.1000genomes.hs37d5²⁷ was used to extract (up to) 500 nt of genomic sequence preceding and succeeding the annotated-donor (GRCh37)."

R2.6 *"as defined by the 5th percentile for intron length in the human genome"... which is?*

We have refined this sentence and corrected an error (we had written 5th instead of 1st percentile in error).

PG 20, Line 1082:

"(as defined by the 1st percentile for intron length in the human genome = 80 nt)"

R2.7 *"Decoy-donor depletion was calculated using an adapted method" adaptive?*

We have edited this sentence to improve the wording.

PG 21, line 1128:

"Decoy-donor depletion was calculated using a method we adapted from a previous study...."

R2.8 *aligned Rail-RNA -> aligner*

Corrected.

R2.9 *"translated from GRCh38 to GRCh37" urgh come on it's 2021. But I realize it would be a lot of work to redo this.*

We recognise this point and agree it is time to contemporise to GRCh38. However, to explain: our discovery of the predictive utility of splice-junction data in 40K-RNA occurred >2 years into a PhD study, with all other data and analysis founded on the GRCh37 genome assembly. We considered conversion to the GRCh38 genome assembly, but as this would not alter the key learnings, and with funding and time constraints trying to complete the PhD project with published outcomes, whilst managing ongoing COVID lockdowns for our laboratory, we elected to proceed to manuscript submission using GRCh37 datasets.

However, since manuscript submission, we have expanded our method using 300,000 publicly available RNA-Seq files annotated in GRCh38 (300K-RNA). Importantly, we confirm our method of ranking the Top 4 events reliably predicts the nature of multiple forms of variant-associated mis-splicing (exon-skipping, cryptic splice-site activation) with 90% sensitivity - for ~100 clinical cases subject to RNA Diagnostics (donor and acceptor variants). These data will be submitted separately for publication (soon), and the GRCh38 300K-RNA will be made publicly available as a web resource on publication of this second manuscript.

R2.10 *"Counter-intuitively, less than half of activated cryptic..." This is only counter-intuitive to me if the canonical SD is now not used at all in favor of the cryptic SD, but I don't think that is the case typically here? If the cryptic SD is close in strength to the canonical it is intuitive to me the cryptic will be chosen some proportion of the time. Splicing is somewhat noisy after all.*

We agree that this word is unnecessary and have changed the sentence:

PG 8, Line 252

"Intuitively, for most CM-variants the cryptic is strengthened by the variant (Figure 3Bi, orange, example shown in Figure 3Bii). However, less than half of activated cryptics are stronger than the annotated-donor (Figure 3Biii)."

R2.11 *"85-99% weaker" There is an awkward question of scale here. What does splice site strength mean? I'm not sure there's a great answer since it will depend on the specific algorithm. e.g. for spliceAI it means "how probable is this to be an annotated SD", but MES takes a more enrichment based approach. Equivalently one could think of considering the log odds instead of the probability... 90% would then mean something different. I don't think there is a good answer (no fault of the authors) but it might be worth discussing a bit.*

In the particular sentence quoted we were actually referring to the proportion of variants that caused a decrease in score for that donor (across the four algorithms). We have amended the text to clarify:

PG 10, Line 356

"In summary, four independent algorithms concur that cryptic-donor activation typically occurs in response to weakening of the annotated-donor (85 - 99 % of variants)..."

However, we do appreciate the reviewers point that the comparison of predicted splice-site strengths is complex. For that reason, we elected to simply compare the scores given by a specific algorithm on a linear scale for the purpose of summarising key insights, while also providing plots in the supplementary materials reporting the full dataset. We've made more explicit reference to this in the text and legend for figure 3.

PG 8, Line 240:

“The four algorithms use different methods to measure the intrinsic ‘strength’ of a given donor splice-site. In the following discussion we use the term ‘stronger’ and ‘weaker’ to denote a donor that has a higher or weaker score, respectively, according to that algorithm. Comparisons such as ‘weaker by >50%’ simply denote that the score given to the donor has been reduced by more than half by the variant.”

PG 9, Line 278:

“Assessment of algorithmic scores of splice-site strength for AM-variants. Categories such as ‘weaker by >50%’ are assigned based on how the score has been impacted by the variant (i.e. more than halved).”

R2.12 *“NNS, MES, and DF rank the decoy-donor at +57 as the most” ... it is not surprising to me that NNS and MES get this wrong since they don't know (or even see) the G-rich SRE. But it is surprising to me that DF gets this wrong: shouldn't +57 not be used in the RNA-seq database following your logic?*

We suspect in this instance R2 has confused Donor Frequency (DF) with 40K-RNA (previously called Splice Competent Events (SCE)). We hope our emended terminology partly addresses this confusion and we have further refined the next sentence to improve clarity:

PG 14, Line 608:

*“NNS, MES, and DF rank the decoy-donor at +57 as the **strongest** donor - however this donor is enveloped within a G-repeat rich region which may mask it, **and accordingly is not present in 40K-RNA.**”*

Discussion:

R2.13 *“determining spliceosomal ‘usability’ of a decoy splice-site, including 2-4 above.”. I agree SpliceAI models your points 3 and 4, but not 2. When it makes a prediction it doesn't know anything about where the canonical SD is.*

Agreed. We have removed the phrase ‘including 2-4 above’ from the sentence (line 608).

R2.14 *I recommend acceptance on the condition the authors make their literature curated set of known (variant, canonical SD, cryptic SD) publicly available, since this will be a nice resource for the community.*

We agree and have provided the ‘cryptic-donor database’ (annotated-, cryptic- and decoy-donors, with predicted scores from all 4 algorithms for REF and VAR sequences) in the source data.

R2.15 *Oh a final point: I don't like the title. Donor -> Splice donor, and replace the numbers with just “high”. The precise numbers are too dependent on many factors and not particularly meaningful in themselves.*

This point was also raised by R3. We agree the title could be improved, though feel the term 'high' is too vague. We have therefore modified our title to incorporate collective feedback from our reviewers:

“90% of variant-activated cryptic-donors are also (mis)used stochastically in population-based RNA-Seq data.”

Reviewer #3 (Remarks to the Author):

R3.1.1 *The title is misleading, the approach proposed by the authors is not general and can not be applied for the "prediction of variant-associated cryptic-donors" as it is. It is my understanding that RNA-sequencing data are required, which are not always available in clinical studies.*

R3.1.2 *The authors claim that their method can be used to evaluate and understand the pathological potential of variants currently considered to be VUS, however this is not the case, since the approach they propose can be used only for a limited selection of cases.*

We cannot be sure, though suspect R3 may have misinterpreted our method. Based on several comments, we feel it is possible R3 has interpreted our method as an analysis of RNA-Seq from affected individuals. In which case, we can see the basis for this comment, and agree clinical RNA-Seq is not yet widely available.

To clarify, this manuscript describes an evidence-based method to reliably predict cryptic splice-site activation for individuals for whom RNA assay data is not available. Our 40K-RNA web-resource of splice-junctions compiled from 40,233 publicly-available RNA-Seq samples reports and ranks the most commonly (mis)used cryptic donors proximal to every ensembl annotated-donor.

See response R1.5 and the new conclusion statement included in the discussion PG 18, Line 1005:

“Our research establishes that for AM-variants, if a cryptic-donor is activated, in 87% of cases it will be among the top 4 events.”

In response to R3.1.1, please see our response to R2.15 and our emended title, which hopefully also mitigate confusion on the applicability of our study:

“90% of variant-activated cryptic-donors are also (mis)used stochastically in population-based RNA-Seq data.”

R3.2.1 *The evaluation of the sensitivity and specificity of the approach proposed by the authors is not carried out in a rigorous manner. No cross fold validation, training and/or testing sets are used to provide an unbiased estimate.*

R3.2.2 *No independent dataset is used (for example one based on long read sequencing technologies).*

R3.2.3 *No wet-lab validation is performed.*

R3.2.4 *With the paper in the current form authors can not claim that they have 87% sensitivity and 95% specificity.*

1. We respectfully disagree that our analyses were not rigorous. Cross-fold validation, and training/testing sets are used for machine-learning approaches. Our method is empirical and not based on machine-learning.
2. 40K-RNA is derived from public RNA-seq data, is independent of the database of cryptic donors, and does not present any bias.
3. The cryptic donor variants have undergone wet-lab validation as a pre-requisite to being included in the study. Additionally, all 40K-RNA is derived entirely from wet-lab experimentation (i.e. RNA-seq on 40,233 samples)
4. With empirical (i.e. evidence-based) data, it is appropriate to report sensitivity and specificity using our large dataset of confirmed variant-activated cryptic donors, as one would for evaluation of any clinical test.

R3.3 *The method for the prediction of cryptic donor sites is based on the integration of a pre-existing tool SpliceAI with RNA-sequencing data. Per se it does not provide the required level of scientific advancement/novelty required for a top-level scientific publication.*

We are not quite sure how to respond, other than to say that we respectfully disagree. Being able to reliably predict the likely nature of variant-associated mis-splicing (based on publicly available and independently verifiable evidence) is a critical gap in clinical genetics and an important finding.

We cannot be sure, though feel the contrasting opinion of R3 compared to R1 & R2 *may* be due to a mis-understanding of our method, and therefore, the implication of the findings – which tells us we need to improve clarity in our writing and nomenclature, and we have tried to do this.

R3.4 *Additionally, levels of sensitivity depend dramatically on depth of sequencing of the RNAseq.*

We agree. Ultra-deep sequencing and enrichment protocols will substantially improve the splice-junction dataset for disease genes with lower expression levels. We acknowledge this important point in the results and discussion, and to address this comment, have added extra emphasis:

PG 16, line 869:

“...only 29% of cryptic-donors are detected in 40K-RNA in transcripts where normal splicing had a maximum read count of < 100 (Figure S6C). Consequently, we assessed SAI as a complementary approach for situations where our empirical method is underpowered or not well suited.”

R3.5 *Finally, it is the opinion of this reviewer that availability of RNA sequencing data is not a common feature in several clinical studies.*

Refer to our response to R3.1

R3.6 *Additionally it not clear why the authors decided to use SpliceAI and not any other tool/tools that they considered in the study.*

Specificity and sensitivity data for NNS, MES and DF was presented in Figure S6A, which we used to decide on the most useful tool moving forward. We have added a sentence which outlines this explicitly.

PG 16, line 875:

“SAI showed higher sensitivity than NNS, and comparable sensitivity to MES and DF, while greatly improving on their specificity (Figure S6A).”

R3.7 *Most of the analyses are purely descriptive (as in they present descriptive statistics about splicing donor sites), the novel approach proposed by the authors Donor Frequency (DF), simply captures the nucleotide composition around splice donor sites and is not substantially different from any consensus-PWSM (positional weight scoring matrix) based method. As such the analyses based on DF score do not provide interesting additional insights apart from the fact that a specific consensus sequence is preferentially observed.*

We disagree. The Position Weight Matrix (PWM) scoring method is where each nucleotide in the sequence is given a weighting based on the frequency of this base at the analogous position of the consensus splice-site. PWM does not take into account the ability of a specific combination of consecutive nucleotides to mediate competitive binding of successive factors during different stages of spliceosomal assembly for execution of both splicing reactions. While U1 binding is a key step to commit an intron for splicing via formation of the early spliceosome, U1 is competitively displaced by the tri-snRNP complex before either of the two splicing reactions is performed. Thus while a lot of attention is placed on complementarity to U1, there is a delicate balancing act of binding affinities required for U1 to be competitively displaced by the tri-snRNP, with U5 subsequently binding the exonic side of the donor and U6 binding the intronic side of the donor. No-one knows how the helicases manage to transition the donor from U1 to the tri-snRNP (because transitional stages of spliceosome assembly can't be captured by cryoEM). DF captures the diversity of specific sequence combinations able to do this in humans. PWM does not.

R3.8 *Splicing is also mediated/controlled by epigenetic factors and/or regulation, so it would be appropriate to discuss the approach proposed by the user should be at least discussed in this context*

We agree that splicing is controlled by epigenetic factors – both by methylation of DNA and of RNA. In the paper we discuss the impact of regulatory elements (i.e., the intronic G repeats) that appear to mask cryptic splice-sites from use by the spliceosome. We don't feel

adding further discussion about epigenetic modification will be helpful. However, we add more specific discussion explaining why 40K-RNA helps to identify 'usable' cryptic donors that confound many algorithms.

PG 14, Line 616:

“Whether or not a cryptic donor can be used is influenced by a constellation of features: the consensus donor sequence, as well as proximal and more distal splicing regulatory elements. Regulatory elements are not factored by many algorithms, though may be identified by SAI, likely underpinning its enhanced capabilities in recognition of usable (cryptic) splice-sites. In contrast, 40K-RNA uses empirical evidence from RNA-Seq data that reveals which cryptic splice-sites are usable in the context of the specimens tested.”

Minor remarks

R3.9 *Some references are incomplete (see ref. 12 or 14, where the journal is missing). Please check all.*

Thank you for alerting us to this problem - we found this was due to an error in our reference manager. We have corrected this fault in reference auto-formatting and have cross-checked our references are now formatted correctly in this revision.

Reviewers' Comments:

Reviewer #1:

Remarks to the Author:

The authors have addressed all my concerns. I have no further comments.

Reviewer #2:

Remarks to the Author:

I'm satisfied with the author rebuttal and recommend acceptance. My impression is that R3 missed the main point of the paper being using a large cohort of public RNA-seq, rather than RNA-seq for the specific cases under study.

I would like to suggest one more alteration to the title to get splic(ing) into it:

90% of variant-activated cryptic SPLICE donors are also (mis)used stochastically in population-based RNA-Seq data

REVIEWER COMMENTS

Reviewer #2 (Remarks to the Author):

R2.1 *I would like to suggest one more alteration to the title to get splic(ing) into it: 90% of variant-activated cryptic SPLICE donors are also (mis)used stochastically in population-based RNA-Seq data*

In light of this comment and the editor's suggestion we have amended our title to: "Empirical prediction of variant-activated cryptic splice donors using population-based RNA-Seq data"